# Aspirin better than clopidogrel on major adverse cardiovascular events reduction after ischemic stroke: A retrospective nationwide cohort study

**Amelia Nur Vidyanti**[1,2], **Lung Chan**[3,4], **Cheng-Li Lin**[5], **Chih-Hsin Muo**[5], **Chung Y. Hsu**[6,7], **You-Chia Chen**[4,8], **Dean Wu**[4,9], **Chaur-Jong Hu**[3,4,8,10]*

1 International Master/PhD Program in Medicine, College of Medicine, Taipei Medical University, Taipei, Taiwan, 2 Department of Neurology, Faculty of Medicine, Public Health and Nursing, Universitas Gadjah Mada, Yogyakarta, Indonesia, 3 Department of Neurology, College of Medicine, Taipei Medical University, Taipei, Taiwan, 4 Department of Neurology and Stroke Center, Shuang Ho Hospital, Taipei Medical University, New Taipei City, Taiwan, 5 Management Office for Health Data, Clinical Trial Research Center, China Medical University Hospital, Taichung, Taiwan, 6 Department of Neurology, China Medical University Hospital, Taichung, Taiwan, 7 Graduate Institute of Clinical Medical Science, China Medical University, Taichung, Taiwan, 8 Taipei Neuroscience Institute, Taipei, Taiwan, 9 TMU Neuroscience Research Center-Sleep Medicine, Taipei, Taiwan, 10 The PhD Program for Neural Regenerative Medicine, College of Medical Science and Technology, Taipei Medical University, Taipei, Taiwan

* chaurjongh@tmu.edu.tw

**Data Availability Statement:** All relevant data are within the manuscript.

## Abstract

### Background

Several clinical trials reported that clopidogrel was superior to aspirin in secondary stroke prevention by reducing the risk of major adverse cardiovascular events (MACE). We aimed to compare the efficacy of clopidogrel with aspirin in reducing one-year risk of MACE based on real-world evidence from Taiwan Health Insurance Database.

### Methods

We identified ischemic stroke patients between 2000 and 2012 who took aspirin or clopidogrel within 7 days of stroke onset for 1-year follow-up. The primary outcome was one-year MACE including recurrent stroke, acute myocardial infarction, and death. Propensity score matching and conditional Cox proportional hazards regression were conducted to control the confounding factors.

### Results

From 9,089 ischemic stroke patients, we found 654 patients on aspirin and 465 patients on clopidogrel who met the selective inclusion criteria. After propensity score matching, 379 patients were selected from each group. The clopidogrel group had a 1.78-fold MACE risk compared with the aspirin group at one-year follow-up (95% CI = 1.41–2.26, *p*<0.01). The MACE-free rate in the aspirin group was 15.74% higher than in the clopidogrel group at one-year follow-up. Sub-analysis of the three components of MACE showed that clopidogrel conferred higher risk of recurrent stroke (OR 1.43, 95% CI = 1.06–1.92, *p* 0.02) and acute

**Funding:** This study was supported in part by Taiwan Ministry of Health and Welfare Clinical Trial Center (MOHW108-TDU-B-212-133004), China Medical University Hospital, Academia Sinica Stroke Biosignature Project (BM10701010021), MOST Clinical Trial Consortium for Stroke (MOST 108-2321-B-039-003), Tseng-Lien Lin Foundation, Taichung, Taiwan, and Katsuzo and Kiyo Aoshima Memorial Funds, Japan. The funders had no role in study design, data collection and analysis, decision to publish, or preparation of the manuscript.

**Competing interests:** The authors have declared that no competing interests exist.

myocardial infarction (OR 3.72, 95% CI = 1.04–13.3, *p* 0.04), but no different risk of death than that of aspirin.

## Conclusions

Among first-ever ischemic stroke patients, secondary stroke prevention using clopidogrel was associated with higher rates of MACE than aspirin. Aspirin might have better efficacy in secondary stroke prevention and was associated with lower risk of MACE. The real-world evidence raises the need to re-assess the current therapeutic options in secondary stroke prevention applying aspirin vs. clopidogrel.

## Introduction

People with first-ever stroke, especially ischemic stroke have higher risk of recurrence not only within 90 days after the stroke onset but also in life-long follow-up [1, 2]. Secondary prevention strategy is critical and the use of antiplatelet agents for these patients should always be carefully assessed on the safety and efficacy issues. Dual anti-platelets therapy is recently recommended only in early stage for short period and the long-term use of single antiplatelet agent is still the standard therapy for patients with ischemic stroke after acute stage [3]. Therefore, to choose the most appropriate single antiplatelet agent is a very important issue.

Aspirin is the most frequently recommended antiplatelet agent for secondary stroke prevention after ischemic stroke. However, a recent meta-analysis of 6 randomized trials (CAPRIE, ESPS-2, MATCH, CHARISMA, ESPRIT, and PRoFESS) conducted to compare the efficacy and safety of different antiplatelet agent for long-term secondary prevention after non-cardioembolic stroke or TIA demonstrated that clopidogrel and combination of aspirin-dipyridamole have a favorable efficacy and safety than the other antiplatelet agents (aspirin alone or a combination of aspirin-clopidogrel) [4]. Moreover, previous systematic review assessing the effectiveness and safety of thienopyridine derivatives (clopidogrel and ticlopidine) versus aspirin concluded that clopidogrel and ticlopidine were modestly more effective than aspirin for preventing serious vascular events in patients at high risk [5]. Although the two studies show that clopidogrel has favorable efficacy than aspirin, most of the study population involved are from Caucasian ethnicity. Study that compares the efficacy of aspirin and clopidogrel in preventing major adverse cardiovascular events (MACE) after ischemic stroke in Asian population is very limited.

Previous clinical trial, CAPRIE, is the only trial which directly comparing the efficacy of aspirin and clopidogrel in patients at risk of ischemic events. The trial reported that long term administration of clopidogrel had better efficacy than aspirin for reducing a composite vascular events of ischemic stroke, acute myocardial infarction (AMI), or vascular death among all groups of patients with atherosclerotic vascular diseases. However, clopidogrel was not more effective than aspirin among the subgroup of stroke patients [6]. Therefore, the comparison of efficacy between these two antiplatelets for secondary stroke prevention remains open for discussion.

Major adverse cardiovascular events (MACE) is a composite outcome of recurrent stroke, AMI, and all cause of death. The risk of MACE after first-ever stroke can be reduced by appropriate antiplatelet agents. The recent meta-analysis study about comparison of efficacy between some antiplatelets in secondary stroke prevention aforementioned above derived from six large clinical trials [4]. Noteworthy that in clinical trials it might be impossible to recruit the patients with extremely old age, and high severity of vascular diseases with complex

vascular comorbidities. For these reasons, about 75% of stroke patients from clinical practice (real-world practice) could not be enrolled based on the selective inclusion/exclusion criteria in those randomized-controlled trials [7, 8]. The limitation of those trials with restricted patient populations raises the need to explore real-world evidence to assess the safety and efficacy of aspirin vs. clopidogrel in the large patient population who have been prescribed aspirin or clopidogrel. By searching real-world evidence, the safety and efficacy of antiplatelet agents can be more reliably measured. In this study, we aimed to evaluate the efficacy of clopidogrel compared to aspirin by calculating the MACE risk between the two groups based on real-world evidence from the National Health Insurance Research Database (NHIRD) in Taiwan.

## Materials and methods

### Study design and population

We used Taiwan's NHIRD for this retrospective cohort study. NHIRD contained one million insured patients randomly selected from the 2000-Registry for Beneficiaries in a National Health Insurance Program which has been established by the National Health Insurance Administration, Ministry of Health and Welfare in Taiwan. The data include patients' personal characteristics from all outpatient and inpatient medical services in Taiwan.

We identified 35.396 ischemic stroke patients between 2000–2012. There were 9089 patients who were prescribed aspirin or clopidogrel within 7 days after stroke onset. The final number was 1119 which met the inclusion criteria. The inclusion criteria: A. Ischemic stroke patients admitted in 2000–2012; B. Received aspirin 100 mg/day or clopidogrel 75 mg/day within 7 days of stroke onset. The exclusion criteria: Age <20 years; treatment with combination of aspirin and dipyridamole, cilostazol, ticlopidine, or warfarin during hospitalization or within 7 days of stroke onset; both aspirin and clopidogrel were prescribed within 7 days of stroke onset; with any antithrombotic (including aspirin, clopidogrel, combination of aspirin and dipyridamole, cilostazol, ticlopidine, and warfarin) before stroke admission; with aspirin use < 6 month within one year after discharge; with clopidogrel use < 6 month within one year after discharge. This study has received ethical approval from Taipei Medical University-Joint Institutional Review Board (TMU-JIRB 057/20140202). Informed consent was waived owing to the retrospective nature of the study.

### Propensity score matching

For reducing the selection bias, we used propensity score (PS) matching analysis for selected comparisons. The algorithm of propensity score matching used "best" matched subjects with the highest digit (8 digits) sequentially to the lowest digits (1 digit). If a case does not have a matched control, it is removed from this study.

### Outcome

The primary outcome was MACE including recurrent stroke, AMI, and all cause of death which occurred within one year after stroke admission. All study subjects were followed from the stroke admission until MACE occurred. Those without MACE were followed until one year after stroke admission, or were withdrawn from the program.

### Statistical analysis

Chi-square test was used to test the different of gender and baseline comorbidities between the aspirin and clopidogrel groups. Student's t-test was used to measure the age difference Wilcoxon Rank sum test was used to test the different of hospitalization days. Incidence of MACE

was the sum of MACE occurred divided by the sum of follow-up periods between the two groups. Cox proportional hazard model was used to estimate the MACE risk before the PS matching. After matching, we used Cox proportional hazard model to adjust matched pairs. Cumulative incidence of MACE was plotted by Kaplan-Meier analysis. All statistical analyses were assessed by SAS software Version 9.4 (SAS Institute Inc., Cary, NC, USA). The significant level was set with $p < 0.05$ in two-tailed tests.

## Results

### Before PS matching

We collected 1119 ischemic stroke patients including 654 patients on aspirin and 465 on clopidogrel. Compared with the aspirin group, the clopidogrel group was older on age (69.9 ± 11.8 vs. 66.7 ± 12.1 years, $p < 0.001$), had more hospitalization days (median 9 vs. 8 days, $p < 0.001$), and higher incidence of hypertension (84.7% vs. 77.1%, $p$ 0.002), dyslipidemia (51.8% vs. 40.3%, $p < 0.001$), heart disease (60.4% vs. 50.5%, $p$ 0.001), and ulcer (62.2% vs. 39.2%, $p < 0.001$) (Table 1). During one-year follow-up, there were 178 (27.3%) and 213 (45.8%) patients with MACE in the aspirin and clopidogrel groups (Table 2). The MACE-free rate in the aspirin group was 11.98% higher than that in the clopidogrel group at one-year follow-up (Fig 1A). Compared with the aspirin group, the clopidogrel group had a 2.05-fold MACE risk (95% CI = 1.68–2.50), 1.69-fold risk of recurrent stroke (95% CI = 1.32–2.17), and a 3.41-fold risk of AMI (95% CI = 1.21–9.68). For the safety profile, clopidogrel group had a 2.52-fold risk (95% CI = 1.88–3.39) of gastrointestinal (GI) bleeding than aspirin group (Table 2).

**Table 1. Baseline Characteristics before and after Propensity Score (PS) Matching.**

| Characteristic | Before PS matching | | | After PS matching[a] | | |
|---|---|---|---|---|---|---|
| | Aspirin (N = 654) | Clopidogrel (N = 465) | *p* value | Aspirin (N = 379) | Clopidogrel (N = 379) | *p* value |
| Age — year; means ± SD | 66.7 (12.1) | 69.9 (11.8) | <0.001 | 68.9 (11.2) | 68.9 (12.2) | 0.98 |
| Hospitalization days — median (interquartile range) | 8 (7) | 9 (11) | <0.001 | 8 (9) | 9 (10) | 0.11 |
| Male — no. (%) | 372 (57.1) | 257 (55.3) | 0.53 | 198 (52.2) | 217 (57.3) | 0.17 |
| Baseline comorbidity — no. (%) | | | | | | |
| Obesity | 4 (0.61) | 5 (1.08) | 0.50 | 3 (0.79) | 4 (1.06) | 1.00 |
| Hypertension | 502 (77.1) | 394 (84.7) | 0.002 | 308 (81.3) | 314 (82.9) | 0.57 |
| Diabetes mellitus | 253 (38.9) | 202 (43.4) | 0.13 | 168 (44.3) | 161 (42.5) | 0.61 |
| Dyslipidemia | 262 (40.3) | 241 (51.8) | <0.001 | 192 (50.7) | 183 (48.3) | 0.51 |
| Previous CVA / TIA | 68 (10.5) | 40 (8.60) | 0.30 | 45 (11.9) | 35 (9.23) | 0.24 |
| Heart disease | 329 (50.5) | 281 (60.4) | 0.001 | 212 (55.9) | 215 (56.7) | 0.83 |
| Atrial fibrillation | 25 (3.84) | 24 (5.16) | 0.29 | 16 (4.22) | 22 (5.80) | 0.32 |
| Ischemic heart | 228 (35.0) | 219 (47.1) | <0.001 | 159 (42.0) | 165 (43.5) | 0.66 |
| Heart failure | 52 (8.00) | 69 (14.8) | <0.001 | 41 (10.8) | 49 (12.9) | 0.37 |
| Smoking | 5 (0.77) | 9 (1.94) | 0.08 | 4 (1.06) | 9 (2.37) | 0.26 |
| Alcohol use | 24 (3.69) | 27 (5.81) | 0.09 | 15 (3.96) | 21 (5.54) | 0.31 |
| Ulcer | 255 (39.2) | 289 (62.2) | <0.001 | 218 (57.5) | 210 (55.4) | 0.56 |
| Malignancy | 22 (3.38) | 19 (4.09) | 0.54 | 12 (3.17) | 17 (4.49) | 0.34 |

[a]Before PS matching, the baseline characteristics between two groups were significantly different (p < 0.05) due to possible sampling bias (population ratio: aspirin/clopidogrel = 5.05/1). After PS matching, there were no significant differences among two groups for any variables.

PS, propensity score; CVA, cerebrovascular accident; TIA, transient ischemic attack.

**Table 2. Major adverse cardiovascular events, intracerebral haemorrhage, and GI bleeding before and after PS matching.**

| | Before PS Matching | | | | After PS Matching | | | |
|---|---|---|---|---|---|---|---|---|
| | Aspirin (N = 651) | Clopidogrel (N = 465) | HR[a] (95% CI) | p value | Aspirin (N = 379) | Clopidogrel (N = 379) | HR[b] (95% CI) | p value |
| Major adverse cardiovascular events within 12 months after stroke onset | 178 (27.3) | 213 (45.8) | 2.05 (1.68–2.50) | <0.001 | 115 (30.3) | 172 (45.4) | 1.78 (1.41–2.26) | <0.001 |
| Recurrent stroke (any type) | 117 (18.0) | 132 (28.4) | 1.69 (1.32–2.17) | <0.0001 | 77 (20.3) | 104 (27.4) | 1.43 (1.06–1.92) | 0.02 |
| AMI | 5 (0.77) | 12 (2.58) | 3.41 (1.21–9.68) | 0.02 | 3 (0.79) | 11 (2.90) | 3.72 (1.04–13.3) | 0.04 |
| Death (all cause of death) | 8 (1.23) | 9 (1.94) | 1.59 (0.61–4.12) | 0.34 | 3 (0.79) | 7 (1.85) | 2.53 (0.61–9.09) | 0.22 |
| Intracerebral hemorrhage | 18 (2.76) | 17 (3.66) | 1.33 (0.69–2.58) | 0.40 | 8 (2.11) | 15 (3.96) | 1.89 (0.79–4.51) | 0.15 |
| GI bleeding | 71 (10.9) | 115 (24.7) | 2.52 (1.88–3.39) | <0.001 | 44 (11.6) | 102 (26.9) | 2.60 (2.82–3.70) | <0.001 |

PS, propensity score; HR, hazard ratio; CI, confidence interval; GI, gastrointestinal.

[a]Unadjusted HR.

[b]Adjusted HR for matched pairs.

## After PS matching

There were 379 ischemic stroke patients on clopidogrel and 379 matched-patients on aspirin. There were comparable distributions of age, gender, hospitalization days, and baseline comorbidity between the two groups (Table 1). During one-year follow-up, there were 115 (30.3%) and 172 (45.4%) patients with MACE in aspirin and clopidogrel groups respectively (Table 2). The MACE-free rate in the aspirin group was 15.74% higher than in the clopidogrel group at one-year follow-up (Fig 1B). Compared with the aspirin group, the clopidogrel group had a 1.78-fold MACE risk (95% CI = 1.41–2.26), 1.43-fold risk of recurrent stroke (95% CI = 1.06–1.92), and a 3.72-fold risk of AMI (95% CI = 1.04–13.3) (Table 2). For the safety profile, clopidogrel group had a 2.60-fold risk (95% CI = 2.82–3.70) of gastrointestinal (GI) bleeding than

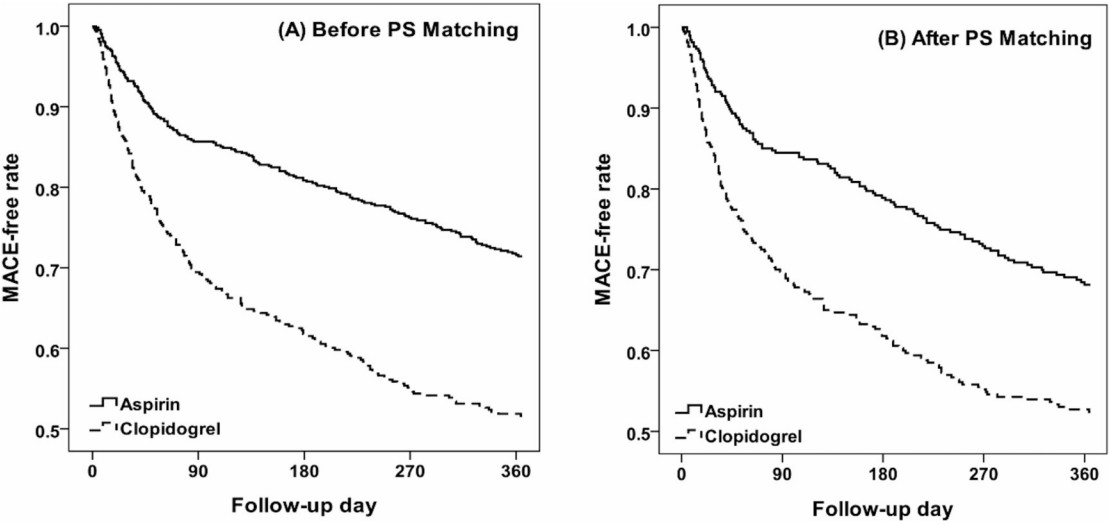

**Fig 1.** MACE-free rate between aspirin and clopidogrel groups before (A) and after (B) PS matching.

aspirin group. However, the risks of intracerebral haemorrhage and death were not different between groups before and after matching.

## Discussion

The present study shows the MACE rate was significantly higher in the clopidogrel group than the aspirin group after one-year follow-up. Patients on clopidogrel had higher risk of recurrent stroke and AMI than those on aspirin. However, the risk of death was similar either in patients on clopidogrel or aspirin.

The higher MACE rate in the clopidogrel group was unexpected. The CAPRIE trial reported that patients treated with clopidogrel had lower risk of composite vascular events (ischemic stroke, AMI, or death) than aspirin (5.32% vs 5.83%), with a relative risk reduction (RRR) of 8.7% in favor of clopidogrel (95% CI = 0.3–16.5, *p* 0.043). However, this finding might be due to the clopidogrel action which conferred the most significant efficacy mainly in the subgroup of patients with peripheral artery disease (PAD) with relative risk reduction (RRR) of 23.8% (95% CI = 8.9–36.2, P<0.01), but not in subgroups of stroke and AMI [6]. Approximately 33% patients with PAD were recruited in the CAPRIE trial to raise the possibility that clopidogrel appeared to be of greater benefit than aspirin in the overall results. Although the number of patients with PAD was not well documented in the present study, it should be low since the prevalence of PAD is much lower in Asian than in Western countries [9]. The prevalence of PAD in Taiwan is approximately 7.2% as previously reported [10].

The reasons why aspirin could have higher efficacy than clopidogrel in the present study could be multi-factorial. First, there is a possibility for clopidogrel resistance in our study subject. Antiplatelet resistance in ischemic stroke patients has emerged as a focus of interest in recent years [11]. Both clopidogrel and aspirin could have a resistance or non-responsiveness causing treatment failure in ischemic stroke patients [11, 12]. Aspirin resistance could be related to genetic polymorphisms of COX-1, COX-2 or thromboxane A2 synthase, while clopidogrel resistance could be due to polymorphisms of receptor P2Y12, enzymes CYP3A4, CYP1A2, CYP2C19, or ABCB1 [11]. There has been no prior study to compare the genetic polymorphisms of aspirin in Asian and Caucasian population. However, the genetic polymorphisms of clopidogrel in Asian, particularly in Chinese population, is reported to be higher than Caucasian. Carriers of the CYP2C19 loss-of-function allele account for 50% to 60% of Chinese population, while they account for 15% to 30% of Caucasian [13–16]. Carriers of this allele will have poor metabolic rate of clopidogrel. This might confer higher risk of clopidogrel resistance in our study subject. A recent meta-analysis study reported that the prevalence of high on-treatment platelet reactivity (HTPR), which is associated with the efficacy of antiplatelet therapy in patients with ischemic stroke or TIA, on aspirin was 23% (95% CI: 20–28%) and on clopidogrel was 27% (95% CI: 22–32%) [17]. It shows that aspirin might be a little more effective than clopidogrel for antiplatelet therapy in secondary stroke prevention.

Furthermore, in addition to its antiplatelet properties, aspirin also exhibits pharmacological benefit in reducing vascular injury by its antioxidant effect via inhibition of lipid peroxidation and DNA damage to reduce the generation of $^*$OH free radicals [18]. Aspirin also has anti-inflammatory effects by inhibiting cyclooxygenase to lessen the progression of vascular injury in patients with cardiovascular and cerebrovascular diseases [19].

In the present study, the MACE rates for aspirin and clopidogrel are 30.3% and 45.4% respectively. They are substantially higher than those reported in previous studies showing the MACE rates around 10% for both antiplatelet agents [20, 21]. The study of Lee et al (2014) also reported higher rates of MACE, it was 23.7% for clopidogrel and 38% for aspirin [22]. Although this study also used the same database of Taiwan NHID, the inclusion criteria was

totally different as the objective was to compare clopidogrel initiation vs aspirin re-initiation for vascular risk reduction among patients with ischemic stroke and aspirin resistance. Hence, the result of this study favored clopidogrel than aspirin re-initiation for reducing the MACE rates [22].

Another study using Taiwan Stroke Registry (TSR) found different results. Although this study applying TSR did not measure the MACE rate, the recurrent stroke rates were quite low, 3.46% for aspirin and 3.79% for clopidogrel [23]. TSR program is a government-funded project of 64 stroke centers in academic and community hospitals in Taiwan (most tertiary hospitals). TSR data has been acknowledged as representative of the national stroke population in the NHIRD [23]. However, NHIRD actually covers the population of outpatient and inpatient services from the local and primary hospitals around Taiwan, but not only tertiary ones [22]. Hence, the characteristics and outcomes of the patients between these two databases might be different. Nevertheless, the reason for the higher rates of MACE in the present study remains to be further explored with more detailed analyses of big data bases between NHIRD and TSR.

Another issue in the present study was about the safety profile of aspirin and clopidogrel. Although the risk of intracerebral haemorrhage was similar between the two groups, GI bleeding was more commonly found in patients on clopidogrel than those on aspirin. It showed contradictory with previous studies [6, 24]. Yet, we could not address the reasons behind this finding. This finding, altogether with the higher rates of MACE among clopidogrel group in the present study, might reflect the gaps between the real-world practice and clinical trials.

In the present study, we used real-world data derived from real-world practice that are more in accord with the real-world condition, not under the more stringent selection of the study population in randomized controlled trials (RCTs). Therefore, the efficacy and safety of antiplatelets might yield different results. Real-world data and real-world evidence (RWE) are increasingly recognized to be of value in health care decisions. USA Food and Drug Administration (FDA) uses real-world data and RWE to monitor post-market safety and adverse events of drugs as well as to make regulatory decisions. Medical product developers also use real-world data and RWE to support clinical trials and observational studies to produce innovative new treatment approaches [25].

Due to its effectiveness, low cost and availability worldwide, aspirin is still recommended as the first-line antiplatelet agent for secondary stroke prevention [3]. However, the pharmacoeconomic analysis of aspirin versus clopidogrel thus far in favor of clopidogrel based on the data derived from the CAPRIE trial [26, 27]. The cost-effectiveness of both drugs should also be further redefined by using RWE. Based on our findings in addition to the notion that aspirin has much lower cost than clopidogrel, we suggest that the efficacy of aspirin is superior to clopidogrel for ischemic stroke on MACE prevention. Results derived from the present study may be of value to assist clinicians in making decision to choose the more appropriate antiplatelet agent for better longer-term outcome.

There are several limitations in the present study. First, this is a retrospective cohort study with the clinical decision for choosing aspirin or clopidogrel were not well documented. Second, even with PS matching, inherent biases might still exist including confounding factors associated with highly variable health profiles and concurrent use of a variety of medications other than antiplatelet agents. Those variables might affect the safety and efficacy of antiplatelets agent and proclivity for developing MACE. Third, the findings derived from the present study may not be applicable to other races since this study was mostly on ethnic Chinese patients. Fourth, there were no adverse events available from the database to enrich the safety profile of both antiplatelets used. Hence, this issue could not be discussed further. Finally, the present study covers a follow-up period of only 1 year for determining the risk of MACE. Follow-up for longer period is likely to strengthen the apparent difference in long-term risk of

developing MACE between 2 patient populations taking aspirin and clopidogrel for secondary stroke prevention.

## Conclusions

Compared with clopidogrel, aspirin was associated with a reduced risk of MACE at one-year follow up among ischemic stroke patients. This real-world evidence from Taiwan NHIRD raises the need to re-assess the current therapeutic options related to antiplatelet agents used in secondary stroke prevention.

## Acknowledgments

This study is based on data provided by the Bureau of National Health Insurance of Taiwan's Ministry of Health and Welfare, and managed by the National Health Research Institutes. The authors' interpretations and conclusions do not represent those of the Bureau of National Health Insurance, the Ministry of Health and Welfare, or the National Health Research Institutes.

## Author Contributions

**Conceptualization:** Chaur-Jong Hu.

**Data curation:** Chih-Hsin Muo.

**Formal analysis:** Chih-Hsin Muo.

**Methodology:** Cheng-Li Lin, Chung Y. Hsu, Chaur-Jong Hu.

**Project administration:** Lung Chan, You-Chia Chen.

**Software:** Chih-Hsin Muo.

**Supervision:** Dean Wu, Chaur-Jong Hu.

**Writing – original draft:** Amelia Nur Vidyanti.

**Writing – review & editing:** Amelia Nur Vidyanti, Lung Chan, Cheng-Li Lin, Chung Y. Hsu, You-Chia Chen, Dean Wu, Chaur-Jong Hu.

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
