## [Decision Letter · Decision Letter 0]

15 Jul 2019

PONE-D-19-17386

Aspirin better than clopidogrel on major adverse cardiovascular events reduction after ischemic stroke: a retrospective nationwide cohort study

PLOS ONE

Dear Dr. Hu,

Thank you for submitting your manuscript to PLOS ONE. After careful consideration, we feel that it has merit but does not fully meet PLOS ONE’s publication criteria as it currently stands. Therefore, we invite you to submit a revised version of the manuscript that addresses the points raised during the review process.

We would appreciate receiving your revised manuscript by Aug 29 2019 11:59PM. To enhance the reproducibility of your results, we recommend that if applicable you deposit your laboratory protocols in protocols.io, where a protocol can be assigned its own identifier (DOI) such that it can be cited independently in the future. For instructions see: http://journals.plos.org/plosone/s/submission-guidelines#loc-laboratory-protocols

We look forward to receiving your revised manuscript.

Kind regards,

Aristeidis H. Katsanos, MD, PhD

Academic Editor

PLOS ONE

Reviewers' comments:

Reviewer's Responses to Questions

**Comments to the Author**

1. Is the manuscript technically sound, and do the data support the conclusions?

Reviewer #1: Yes

Reviewer #2: Partly

2. Has the statistical analysis been performed appropriately and rigorously? 

Reviewer #1: I Don't Know

Reviewer #2: Yes

3. Have the authors made all data underlying the findings in their manuscript fully available?

Reviewer #1: Yes

Reviewer #2: Yes

4. Is the manuscript presented in an intelligible fashion and written in standard English?

Reviewer #1: Yes

Reviewer #2: Yes

5. Review Comments to the Author

Reviewer #1: The authors tackled a very challenging subject of the most appropriate antiplatelet medication in secondary stroke prevention. The choice of the most appropriate antiplatelet agent is an unmet need. The literature is varied and makes the option difficult. The new revised stroke guidelines 2018 assumed that the selection of the antiplatelet agent depends on the clinicians’ preference, risk factors, cost, tolerance, and clinical characteristics.

The present scientific work is a retrospective study, including patients from the Taiwan Health Insurance Database. The authors invested a significant effort to compare aspirin with clopidogrel in preventing recurrent vascular events. This was a very thought provoking topic to investigate.

The study is clearly presented and described. The conclusion supported by the results. Nonetheless, there are several limitations that should be addressed and discussed.

1. The authors aimed to demonstrate that aspirin might have better efficacy than clopidogrel in secondary stroke prevention; however, the previous data does not support this conclusion. The findings are not consensus with the previous mentioned study of Lee et al, where the data derived from the same database. Specifically, the authors may wish to comment on how and why the findings are inconsistent with the previous mentioned study.

2. Moreover, the authors should identify more related literature that clarifies and justifies their findings and the controversies over the secondary antiplatelet treatment in stroke patients. Especially, a very recently published meta-analysis of Greving et al (2019) concluded that clopidogrel and aspirin/dipyridamole combination seemed to be the most appropriate choices with a favorable balance between efficacy and safety. Another previous review of the Cochrane Stroke Group trials register and the Antithrombotic Trialists' database concluded that the thienopyridine derivatives (ticlopidine and clopidogrel) are modestly but significantly more effective than aspirin in preventing serious vascular events in patients at high risk (and specifically in TIA/ischemic stroke patients)(Hankey et al 2000).

3. Additionally, the authors referred that the CAPRIE study failed to show a better efficacy of clopidogrel compared to aspirin; however the CAPRIE trial was not designed to determine whether clopidogrel was superior or equivalent to aspirin among stroke patients.

4. The authors mentioned that diabetic patients have higher rates of high on treatment platelet reactivity (HTPR) on clopidogrel and that could be associated with increased risk of MACE. The authors should include more information about the non-responsiveness of antiplatelets in stroke patients with diabetes mellitus. There are many scientific reports indicating higher rates of non-responsiveness to both antiplatelets (aspirin and clopidogrel) in diabetic patients and higher rates of recurrent strokes. In addition, as the authors referred to genes associated with HTPR on clopidogrel, a recent popular view is the potential relation of genetic factors with the prevalence of HTPR in both antiplatelets. There are many studies in the literature on that subject. It is also noteworthy, the black box warning from the US Food and Drug Administration that recommends patients to be tested for the possibility of reduced efficacy of clopidogrel in individuals carrying the CYP2C19*2 loss-of-function allele.

5. While the study appears to be sound, the language is unclear at some points, making it difficult to follow. There are some minor elements of grammar that would improve the article’s presentation. In the abstract, on page 2, line 38 “in 2000-2012”, the word “between” may work better. In the same page, line 46 “inclusion/exclusion criteria”, the sentence “patients on aspirin or clopidogrel who met the inclusion criteria” may work more effectively. On page 4, line 88 the authors should include references of trials comparing the efficacy of aspirin with clopidogrel. On page 5, line 133 referring to the primary outcome including all cause of hospitalization, the authors may wish to consider stating what cause of hospitalization they meant. On Table 1, there is a missing abbreviation for the CVA. At many points in the article, MACE refers to both cardiovascular and cerebrovascular disease. Lastly, the authors should revise the language in the section of “Conclusions” at page 13 to improve readability.

Reviewer #2: The study conducted by Vidyanti et al. had a retrospective design and aimed to evaluate the effect of clopidogrel when compared to aspirin on a combined clinical outcome – Major Adverse Cardiovascular Events (MACE).

The data provided by the author partially support their conclusion. This is a retrospective cohort study and the results should provide insights for another type of design, if relevant for specific population/ethnicity. The chosen design has limitations regarding biases and confounding variables prevented definite conclusions regarding drug efficacy.

The authors stated that all relevant data are within the manuscript and the statistical analysis was performed properly using Propensity score matching, a technique which minimize differences between two groups. This statistical tool was associated with conditional Cox proportional hazards regression for the same purpose.

The manuscript was written in an intelligible fashion and using standard English.

Comments:

1. How was ischemic stroke defined to qualify as an event in the retrospective cohort?

2. Regarding the inclusion criteria, specify the generic name of the substance (e.g., say “combination of Aspirin and Dipyridamole” instead of “Aggrenox”, “Cilostazol” instead of “Pletaal”). It is better to be consistent, as the other drugs were described using generic names.

3. In the lines 83 and 84 of introduction the authors state that Major Adverse Cerebrovascular Events include death, stroke, acute myocardial infarction, bleeding, symptomatic pulmonary embolism, and cardiovascular hospitalization. The definition of MACE usually varies from study to study.

4. In the Outcome session, the authors chose MACE as they defined previously, but they included all causes of hospitalization instead of cardiovascular hospitalization. Please clarify.

5. The authors defined Major Adverse Cerebrovascular Events as including death, stroke, acute myocardial infarction, bleeding, symptomatic pulmonary embolism and cardiovascular hospitalization. I would suggest a detailed description of each component of MACE that make it possible to classify the events.

6. It should be interesting to see the results of the comparison if we include the three most classical MACE events: 1) any type of stroke, including non-fatal primary intracranial hemorrhage; acute myocardial infarction; or death from any cause, including fatal bleeding. In this way the MACE events still will give us an idea of efficacy and safety. It is possible that the higher MACE rates found in the present study compared to other studies are due to a broader definition regarding MACE outcomes.

7. This analysis, including many types of MACE, might have increased the number of factors not related to atherosclerosis that could explain the outcomes considering what was included; for example, all causes of hospitalization.

8. Regarding the MACE events the authors included all-cause of bleeding. Which exact type(s) of bleeding was identified? It would be interesting to specify different types of bleeding separately as part of the safety profile. For instance, the authors only reported rates of intracerebral hemorrhage in table 2. The authors included all causes of bleeding as major cardiovascular events.

9. Is “recurrent stroke” the same as “recurrent ischemic stroke?” Please specify. The same for death; was it all causes of death or only cardiovascular death? Please specify.

10. There is missing report of adverse events regarding aspirin and clopidogrel groups.

11. In a subgroup analysis of the CAPRIE study, the diabetic population benefits more from clopidogrel compared with non-diabetic (Relative Risk Reduction of 13.1% in favour of clopidogrel versus 8.7% overall). However, the authors proposed that the higher proportion of patients with diabetes in the study might explain the findings.

It is also worth to address that there is evidence against this explanation (see above) and clopidogrel showed even more benefit compared to aspirin in higher risk population as per hypercholesterolemia, diabetes, previous coronary bypass surgery, history of more than 1 ischemic event and multiple vascular beds involvement in CAPRIE study. It is possible that higher prevalence of diabetes in the present study is not what has driven the results in favor to aspirin as raised in the discussion.

12. What was the average period of time that patients were on medication in each group?

13. One limitation is that if patients used aspirin or clopidogrel for less than one year and the MACE was calculated within one year, then it is likely that the patient was not under drug protection/effect anymore at the time that some MACE occurred.

Additional suggestions:

1. Line 95 : “ In this study, we aimed to compare the efficacy of clopidogrel with aspirin with MACE included to assess safety between groups based on real-world evidence form the…”. Did the authors mean “In this study, we aimed to evaluate the efficacy of clopidogrel compared to aspirin with MACE included to assess safety between groups based on real-world evidence...” ?

2. Table 1. Hospitalization days – median (interquartile range). There is no range demonstrated. Please clarify variable used

3. “HPR of clopidogrel could also be associated with increased risk of MACE.” Clarify this sentence. Did the authors mean HPR on clopidogrel treatment…

6. PLOS authors have the option to publish the peer review history of their article (what does this mean?). If published, this will include your full peer review and any attached files.

Reviewer #1: No

Reviewer #2: No

---

## [Author Response · Author response to Decision Letter 0]

31 Jul 2019

We thank the reviewers for the valuable critiques and comments. These really improve very much the quality of this manuscript. We have made a major revision on this manuscript according to the requests of reviewers. In the mean time, we add one co-author because of her contribution to this revision process. Below is our point-to-point responses. 

Response to Reviewer #1

1. The authors aimed to demonstrate that aspirin might have better efficacy than clopidogrel in secondary stroke prevention; however, the previous data does not support this conclusion. The findings are not consensus with the previous mentioned study of Lee et al, where the data derived from the same database. Specifically, the authors may wish to comment on how and why the findings are inconsistent with the previous mentioned study.

Response 1: Although derived from a same database, the study conducted by Lee et al (2014) included patients with different inclusion criteria based on the aim of their study. The aim of their study was to compare clopidogrel initiation vs aspirin re-initiation for vascular risk reduction among patients with ischemic stroke on aspirin at the time of their index stroke (index stroke was defined as the first ischemic stroke during the study period from 2003-2009). Only patients with ischemic stroke who received continuous aspirin treatment >30 days before the index stroke were included in their study cohort. In other words, they included ischemic stroke patients with aspirin-failure treatment, in terms of aspirin-resistant patients. For that reason, the MACE rate in that study was higher in aspirin group than clopidogrel, make it different with our study finding.

Nevertheless, we thank reviewer for the comments for this issue. We realize that the statement written in our manuscript was not fully corrected. Hence, we revise the statement about this issue in the discussion part (line 225):

“The study of Lee et al (2014) also reported higher rates of MACE, it was 23.7% for clopidogrel and 38% for aspirin. Although this study also used the same database of Taiwan NHID, the inclusion criteria was totally different as the objective was to compare clopidogrel initiation vs aspirin re-initiation for vascular risk reduction among patients with ischemic stroke and aspirin resistance. Hence, the result of this study favored clopidogrel than aspirin re-initiation for reducing the MACE rates.”

2. Moreover, the authors should identify more related literature that clarifies and justifies their findings and the controversies over the secondary antiplatelet treatment in stroke patients. Especially, a very recently published meta-analysis of Greving et al (2019) concluded that clopidogrel and aspirin/dipyridamole combination seemed to be the most appropriate choices with a favorable balance between efficacy and safety. Another previous review of the Cochrane Stroke Group trials registers and the Antithrombotic Trialists' database concluded that the thienopyridine derivatives (ticlopidine and clopidogrel) are modestly but significantly more effective than aspirin in preventing serious vascular events in patients at high risk (and specifically in TIA/ischemic stroke patients) (Hankey et al 2000).

Response 2: We thank the reviewer for the suggestions. We have revised the literature review to justify our wording. Here it is the revised statements:

“Aspirin is the most frequently recommended antiplatelet agent for secondary stroke prevention after ischemic stroke. However, a recent meta-analysis of 6 randomized trials (CAPRIE, ESPS-2, MATCH, CHARISMA, ESPRIT, and PRoFESS) conducted to compare the efficacy and safety of different antiplatelet agent for long-term secondary prevention after non-cardioembolic stroke or TIA demonstrated that clopidogrel and combination of aspirin-dipyridamole have a favorable efficacy and safety than the other antiplatelet agents (aspirin alone or a combination of aspirin-clopidogrel) (1). Moreover, previous systematic review assessing the effectiveness and safety of thienopyridine derivatives (clopidogrel and ticlopidine) versus aspirin concluded that clopidogrel and ticlopidine were modestly more effective than aspirin for preventing serious vascular events in patients at high risk (2). Although the two studies show that clopidogrel has favorable efficacy than aspirin, most of the study population involved are from Caucasian ethnicity. Study that compares the efficacy of aspirin and clopidogrel in preventing major adverse cardiovascular events (MACE) after ischemic stroke in Asian population is very limited.”

3. Additionally, the authors referred that the CAPRIE study failed to show a better efficacy of clopidogrel compared to aspirin; However, the CAPRIE trial was not designed to determine whether clopidogrel was superior or equivalent to aspirin among stroke patients.

Response 3: Yes, we agree with that. We realize that CAPRIE was designed to assess relative efficacy of clopidogrel and aspirin in reducing MACE risk among patients not only with ischemic stroke, but also with AMI, or symptomatic PAD. Therefore, we have revised that statement as below:

“Previous clinical trial, CAPRIE, is the only trial which directly comparing the efficacy of aspirin and clopidogrel in patients at risk of ischemic events. The trial reported that long term administration of clopidogrel had better efficacy than aspirin for reducing a composite vascular events of ischemic stroke, acute myocardial infarction (AMI), or vascular death among all groups of patients with atherosclerotic vascular diseases. However, clopidogrel was not more effective than aspirin among the subgroup of stroke patients (3). Therefore, the comparison of efficacy between these two antiplatelets for secondary stroke prevention remains open for discussion.”

4. The authors mentioned that diabetic patients have higher rates of high on treatment platelet reactivity (HTPR) on clopidogrel and that could be associated with increased risk of MACE. The authors should include more information about the non-responsiveness of antiplatelets in stroke patients with diabetes mellitus. There are many scientific reports indicating higher rates of non-responsiveness to both antiplatelets (aspirin and clopidogrel) in diabetic patients and higher rates of recurrent strokes. In addition, as the authors referred to genes associated with HTPR on clopidogrel, a recent popular view is the potential relation of genetic factors with the prevalence of HTPR in both antiplatelets. There are many studies in the literature on that subject. It is also noteworthy, the black box warning from the US Food and Drug Administration that recommends patients to be tested for the possibility of reduced efficacy of clopidogrel in individuals carrying the CYP2C19*2 loss-of-function allele.

Response 4: Yes, we totally agree for that. Diabetic patients could have non-responsiveness to antiplatelets, either aspirin or clopidogrel. However, a recent meta-analysis study in 2017 reported that the prevalence of high on-treatment platelet reactivity (HTPR) in patients with ischemic stroke or TIA on aspirin was 23% (95%CI: 20-28%) and on clopidogrel was 27% (95%CI: 22-32%) (4). It shows that aspirin might be a little more effective than clopidogrel. Nevertheless, we realize our statement about the higher percentage of diabetic patients in our study subjects as one of the reasons why aspirin has higher efficacy than clopidogrel might not be appropriate. Therefore, we have omitted this statement and revised it by giving more information about the non-responsiveness or resistance of antiplatelet in ischemic stroke patients in general, also added the literature about HTPR on aspirin and clopidogrel as mentioned above.

Another reason is, carriers of the CYP2C19 loss-of-function allele account for 59% in the Asians especially Chinese population (compared to around 30% in the Caucasians). This genetic polymorphism has been associated with reduced efficacy of clopidogrel (5). Unfortunately, study about the genetic polymorphism of aspirin in Asian population compared with Caucasians are not well documented. Therefore, the possibility of genetic polymorphism of CYP2C19 in our study subject might also be considered as one of the reasons why in our finding aspirin has more favorable effect in reducing MACE risk than clopidogrel.

Here is the revised version (from line 197):

“The reasons why aspirin could have higher efficacy than clopidogrel in the present study could be multi-factorial. First, there is a possibility for clopidogrel resistance in our study subject. Antiplatelet resistance in ischemic stroke patients has emerged as a focus of interest in recent years (6). Both clopidogrel and aspirin could have a resistance or non-responsiveness causing treatment failure in ischemic stroke patients (6, 7). Aspirin resistance could be related to genetic polymorphisms of COX-1, COX-2 or thromboxane A2 synthase, while clopidogrel resistance could be due to polymorphisms of receptor P2Y12, enzymes CYP3A4, CYP1A2, CYP2C19, or ABCB1 (6). There has been no prior study to compare the genetic polymorphisms of aspirin in Asian and Caucasian population. However, the genetic polymorphisms of clopidogrel in Asian, particularly in Chinese population, is reported to be higher than Caucasian. Carriers of the CYP2C19 loss-of-function allele account for 50% to 60% of Chinese population, while they account for 15% to 30% of Caucasian (5, 8-10). Carriers of this allele will have poor metabolic rate of clopidogrel. This might confer higher risk of clopidogrel resistance in our study subject. A recent meta-analysis study reported that the prevalence of high on-treatment platelet reactivity (HTPR), which is associated with the efficacy of antiplatelet therapy in patients with ischemic stroke or TIA, on aspirin was 23% (95% CI: 20-28%) and on clopidogrel was 27% (95% CI: 22-32%) (4). It shows that aspirin might be a little more effective than clopidogrel for antiplatelet therapy in secondary stroke prevention.”

“Furthermore, in addition to its antiplatelet properties, aspirin also exhibits pharmacological benefit in reducing vascular injury by its antioxidant effect via inhibition of lipid peroxidation and DNA damage to reduce the generation of *OH free radicals (11). Aspirin also has anti-inflammatory effects by inhibiting cyclooxygenase to lessen the progression of vascular injury in patients with cardiovascular and cerebrovascular diseases (12).”

5. While the study appears to be sound, the language is unclear at some points, making it difficult to follow. There are some minor elements of grammar that would improve the article’s presentation. In the abstract, on page 2, line 38 “in 2000-2012”, the word “between” may work better. In the same page, line 46 “inclusion/exclusion criteria”, the sentence “patients on aspirin or clopidogrel who met the inclusion criteria” may work more effectively. On page 4, line 88 the authors should include references of trials comparing the efficacy of aspirin with clopidogrel. On page 5, line 133 referring to the primary outcome including all cause of hospitalization, the authors may wish to consider stating what cause of hospitalization they meant. On Table 1, there is a missing abbreviation for the CVA. At many points in the article, MACE refers to both cardiovascular and cerebrovascular disease. Lastly, the authors should revise the language in the section of “Conclusions” at page 13 to improve readability.

Response 5: 

Thank you for your valuable correction. We have revised the language as you suggest.

Line 38 “in 2000-2012” changed into “between 2000 and 2012”

Line 46 “patients on aspirin or clopidogrel who met the inclusion/exclusion criteria” changed into “patients on aspirin or clopidogrel who met the inclusion criteria”

Line 88: We omitted the statement from line 84 to 88 since we have made revision about the newest literature about antiplatelet comparison in secondary stroke prevention (a recent meta-analysis study). 

Line 133: We apologize for the mistake we made here. The definition of MACE that we used in the analysis was only a composite outcome of recurrent stroke, AMI, and all cause of death. Other than that, we did not use it.

On Table 1, we have already added the abbreviation for CVA in the revised manuscript.

It was our mistake to type an abbreviation for MACE as “major adverse cerebrovascular events” (page 4 line 84). It was a typo and supposed to be “major adverse cardiovascular events”. Thank you for your correction about this terminology.

Conclusion has been revised into: “Compared with clopidogrel, aspirin was associated with a reduced risk of MACE at one-year follow up among ischemic stroke patients. This real-world evidence from Taiwan NHIRD raises the need to re-assess the current therapeutic options related to antiplatelet agents used in secondary stroke prevention.” 

Response to Reviewer #2

1. How was ischemic stroke defined to qualify as an event in the retrospective cohort?

Response 1: We used the classification of ischemic stroke according to the code of diagnosis from ICD-9-CM (The International Classification of Diseases, Ninth Revision, Clinical Modification). Those with the code of ICD-9-CM 433-438 were classified as ischemic stroke. This code is the official system of assigning codes to diagnoses and procedures associated with hospital utilization.

2. Regarding the inclusion criteria, specify the generic name of the substance (e.g., say “combination of Aspirin and Dipyridamole” instead of “Aggrenox”, “Cilostazol” instead of “Pletaal”). It is better to be consistent, as the other drugs were described using generic names.

Response 2: Thank you for your valuable suggestions. We have revised the generic name of the drugs. Here is the revised version:

“….; treatment with a combination of aspirin and dipyridamole, cilostazol,…..”

3. In the lines 83 and 84 of introduction the authors state that Major Adverse Cerebrovascular Events include death, stroke, acute myocardial infarction, bleeding, symptomatic pulmonary embolism, and cardiovascular hospitalization. The definition of MACE usually varies from study to study.

Response 3: We apologize for the mistake we made. The terminology of MACE in line 83 is supposed to be Major Adverse Cardiovascular Events, not Cerebrovascular. It was a mistake. In addition to that, we have re-checked our definition of MACE and the data we have. The definition of MACE in our study only consists of recurrent stroke, AMI, and all cause of death (including major bleeding). We did not include those with symptomatic pulmonary embolism. Cardiovascular hospitalization was only limited to those with AMI. Hence, we have revised the statement regarding the definition of MACE into (line 83): “Major adverse cardiovascular events (MACE) is a composite outcome of all cause of death, any type of recurrent stroke, and AMI”

4. In the Outcome session, the authors chose MACE as they defined previously, but they included all causes of hospitalization instead of cardiovascular hospitalization. Please clarify.

Response 4: We thank the reviewer for reminding this. As we answered the critiques of reviewer 1 that it was our mistake about our previous definition of MACE. Here is the revised version:

“The primary outcome was MACE including recurrent stroke, acute myocardial infarction (AMI), and all cause of death which occurred within one year after stroke admission”

5. The authors defined Major Adverse Cerebrovascular Events as including death, stroke, acute myocardial infarction, bleeding, symptomatic pulmonary embolism and cardiovascular hospitalization. I would suggest a detailed description of each component of MACE that make it possible to classify the events.

Response 5: Death here refers to all cause of death since we are not able to identify the real causes of death for every one patient. Recurrent stroke refers to any types of recurrent stroke (ischemic and/or hemorrhagic). 

6. It should be interesting to see the results of the comparison if we include the three most classical MACE events: 1) any type of stroke, including non-fatal primary intracranial hemorrhage; acute myocardial infarction; or death from any cause, including fatal bleeding. In this way the MACE events still will give us an idea of efficacy and safety. It is possible that the higher MACE rates found in the present study compared to other studies are due to a broader definition regarding MACE outcomes.

Response 6: Thank you for your suggestions. This is a great idea. Actually, the data we presented in the original manuscript already included these three efficacy-related MACE. We apologize for leading you to misinterprete due to our mistake in the MACE definition. We have revised the definition of MACE and added the sub-analysis of each three components of MACE as below. Regarding the safety-related outcomes, we also have listed both the risks of intracranial hemorrhage and gastrointestinal bleeding but we do not have the data of fatal bleeding. 

Table 2. Major adverse cardiovascular events, intracerebral haemorrhage, and GI bleeding before and after PS matching.

 Before PS Matching After PS Matching

 Aspirin

(N = 651) Clopidogrel

(N = 465) HRa (95% CI) p value Aspirin

(N = 379) Clopidogrel

(N = 379) HRb (95% CI) p value

Major adverse cardiovascular events within 12 months after stroke onset 178 (27.3) 213 (45.8) 2.05 (1.68-2.50) <0.001 115 (30.3) 172 (45.4) 1.78 (1.41-2.26) <0.001

Recurrent stroke (any type) 117 (18.0) 132 (28.4) 1.69 (1.32-2.17) <0.0001 77 (20.3) 104 (27.4) 1.43 (1.06-1.92) 0.02

AMI 5 (0.77) 12 (2.58) 3.41 (1.21-9.68) 0.02 3 (0.79) 11 (2.90) 3.72 (1.04-13.3) 0.04

Death (all cause of death) 8 (1.23) 9 (1.94) 1.59 (0.61-4.12) 0.34 3 (0.79) 7 (1.85) 2.53 (0.61-9.09) 0.22

Intracerebral hemorrhage 18 (2.76) 17 (3.66) 1.33 (0.69-2.58) 0.40 8 (2.11) 15 (3.96) 1.89 (0.79-4.51) 0.15

GI bleeding 71 (10.9) 115 (24.7) 2.52 (1.88-3.39) <0.001 44 (11.6) 102 (26.9) 2.60 (2.82-3.70) <0.001

PS, propensity score; HR, hazard ratio; CI, confidence interval; GI, gastrointestinal.

aUnadjusted HR, bAdjusted for matched pairs.

7. This analysis, including many types of MACE, might have increased the number of factors not related to atherosclerosis that could explain the outcomes considering what was included; for example, all causes of hospitalization.

Response 7: We agree with the reviewer’s opinion, even after analyzing the outcome of MACE into sub-analysis of recurrent stroke, AMI, and all cause of death, the rates are still higher than previous clinical trials. These differences might reflect the gaps between the real-world practice and clinical trials. 

8. Regarding the MACE events the authors included all-cause of bleeding. Which exact type(s) of bleeding was identified? It would be interesting to specify different types of bleeding separately as part of the safety profile. For instance, the authors only reported rates of intracerebral hemorrhage in table 2. The authors included all causes of bleeding as major cardiovascular events.

Response 8: We have revised table 2 with the additional safety profile consists of the rates of ICH and GI bleeding as shown above. Unexpectedly, the rates of GI bleeding for clopidogrel was higher than aspirin. It shows contradictory with previous studies. Again, this might reflect the gaps between the real-world practice and clinical trials. Yet, we could not address the reasons behind this. We also have listed this point in our discussion.

9. Is “recurrent stroke” the same as “recurrent ischemic stroke?” Please specify. The same for death; was it all causes of death or only cardiovascular death? Please specify.

Response 9: Recurrent stroke is defined as any type of recurrent stroke (ischemic and/or hemorrhagic). The death means all cause of death. We have clarified these definitions in the revised version. 

10. There is missing report of adverse events regarding aspirin and clopidogrel groups.

Response 10: Yes, we agree with that. Unfortunately, we cannot find any adverse events from the database. Therefore, we have listed this issue as one of the limitations in our study.

11. In a subgroup analysis of the CAPRIE study, the diabetic population benefits more from clopidogrel compared with non-diabetic (Relative Risk Reduction of 13.1% in favour of clopidogrel versus 8.7% overall). However, the authors proposed that the higher proportion of patients with diabetes in the study might explain the findings.

It is also worth to address that there is evidence against this explanation (see above) and clopidogrel showed even more benefit compared to aspirin in higher risk population as per hypercholesterolemia, diabetes, previous coronary bypass surgery, history of more than 1 ischemic event and multiple vascular beds involvement in CAPRIE study. It is possible that higher prevalence of diabetes in the present study is not what has driven the results in favor to aspirin as raised in the discussion.

Response 11: We thank the reviewer for this valuable comment. It improves the quality of this study very much. We totally agree with that and we are sorry for the misleading. . Therefore, we have revised the discussion part related to this issue. 

Here is the revised version (from line 197):

“The reasons why aspirin could have higher efficacy than clopidogrel in the present study could be multi-factorial. First, there is a possibility for clopidogrel resistance in our study subject. Anti-platelet resistance in ischemic stroke patients has emerged as a focus of interest in recent years (1). Both clopidogrel and aspirin could have a resistance or non-responsiveness causing treatment failure in ischemic stroke patients (1, 2). Aspirin resistance could be related to genetic polymorphisms of COX-1, COX-2 or thromboxane A2 synthase, while clopidogrel resistance could be due to polymorphisms of receptor P2Y12, enzymes CYP3A4, CYP1A2, CYP2C19, or ABCB1 (1). There has been no prior study to compare the genetic polymorphisms of aspirin in Asian and Caucasian population. However, the genetic polymorphisms of clopidogrel in Asian, particularly in Chinese population, is reported to be higher than Caucasian. Carriers of the CYP2C19 loss-of-function allele account for 50% to 60% of Chinese population, while they account for 15% to 30% of Caucasian (3-6). Carriers of this allele will have poor metabolic rate of clopidogrel. This might confer higher risk of clopidogrel resistance in our study subject. A recent meta-analysis study reported that the prevalence of high on-treatment platelet reactivity (HTPR), which is associated with the efficacy of antiplatelet therapy in patients with ischemic stroke or TIA, on aspirin was 23% (95%CI: 20-28%) and on clopidogrel was 27% (95%CI: 22-32%) (7). It shows that aspirin might be a little more effective than clopidogrel for antiplatelet therapy in secondary stroke prevention.”

“Furthermore, in addition to its antiplatelet properties, aspirin also exhibits pharmacological benefit in reducing vascular injury by its antioxidant effect via inhibition of lipid peroxidation and DNA damage to reduce the generation of *OH free radicals (8). Aspirin also has anti-inflammatory effects by inhibiting cyclooxygenase to lessen the progression of vascular injury in patients with cardiovascular and cerebrovascular diseases (9).”

12. What was the average period of time that patients were on medication in each group?

Response 12: The following are the number of average period of time of drug used from the time of discharge up to one year.

 Mean SD median Q1, Q3

Aspirin 298 69.8 320 238, 358

Clopidogrel 314.8 65.4 340 276, 362

13. One limitation is that if patients used aspirin or clopidogrel for less than one year and the MACE was calculated within one year, then it is likely that the patient was not under drug protection/effect anymore at the time that some MACE occurred.

Response 13: Again, we thank the reviewer for this comment. As the table mentioned above, the anti-platelet treatment covered almost but not all the whole period of MACE analysis, in terms of the whole year. We have listed this critical point into our limitations. Nevertheless, even though the average duration of aspirin use was shorter than that of clopidogrel, the MACE risk in aspirin group was lower than in clopidogrel group. It could enhance the finding that aspirin had better efficacy than clopidogrel.

Additional suggestions:

1. Line 95 : “ In this study, we aimed to compare the efficacy of clopidogrel with aspirin with MACE included to assess safety between groups based on real-world evidence form the…”. Did the authors mean “In this study, we aimed to evaluate the efficacy of clopidogrel compared to aspirin with MACE included to assess safety between groups based on real-world evidence...” ?

Response: Yes, that was what we meant. Thank you for your correction. Here is the revised version:

“In this study, we aimed to evaluate the efficacy of clopidogrel compared to aspirin by calculating the MACE risk between the two groups based on real-world evidence from the National Health Insurance Research Database (NHIRD) in Taiwan.”

2. Table 1. Hospitalization days – median (interquartile range). There is no range demonstrated. Please clarify variable used

Response: Interquartile range (IQR) is calculated by the formula= Q3-Q1; Hence there would be no range needed.

3. “HPR of clopidogrel could also be associated with increased risk of MACE.” Clarify this sentence. Did the authors mean HPR on clopidogrel treatment…

Response: Yes, we meant HPR on clopidogrel treatment. Nevertheless, this statement is no longer available because we have revised the discussion part as we mentioned above (response 11). Thank you for your correction.

Reference:

1. Greving JP, Diener HC, Reitsma JB, Bath PM, Csiba L, Hacke W, et al. Antiplatelet Therapy After Noncardioembolic Stroke. Stroke. 2019;50(7):1812-8.

2. Sudlow CL, Mason G, Maurice JB, Wedderburn CJ, Hankey GJ. Thienopyridine derivatives versus aspirin for preventing stroke and other serious vascular events in high vascular risk patients. Cochrane Database Syst Rev. 2009(4):Cd001246.

3. CAPRIE Steering Committee. A randomised, blinded, trial of clopidogrel versus aspirin in patients at risk of ischaemic events (CAPRIE). The Lancet. 1996;348(9038):1329-39.

4. Fiolaki A, Katsanos AH, Kyritsis AP, Papadaki S, Kosmidou M, Moschonas IC, et al. High on treatment platelet reactivity to aspirin and clopidogrel in ischemic stroke: a systematic review and meta-analysis. Journal of the neurological sciences. 2017;376:112-6.

5. Pan Y, Chen W, Xu Y, Yi X, Han Y, Yang Q, et al. Genetic polymorphisms and clopidogrel efficacy for acute ischemic stroke or transient ischemic attack: a systematic review and meta-analysis. Circulation. 2017;135(1):21-33.

6. Topçuoglu MA, Arsava EM, Ay H. Antiplatelet resistance in stroke. Expert review of neurotherapeutics. 2011;11(2):251-63.

7. Mijajlovic M, Shulga O, Bloch S, Covickovic‐Sternic N, Aleksic V, Bornstein N. Clinical consequences of aspirin and clopidogrel resistance: an overview. Acta Neurologica Scandinavica. 2013;128(4):213-9.

8. Desta Z, Zhao X, Shin J-G, Flockhart DA. Clinical significance of the cytochrome P450 2C19 genetic polymorphism. Clinical pharmacokinetics. 2002;41(12):913-58.

9. Mega JL, Close SL, Wiviott SD, Shen L, Hockett RD, Brandt JT, et al. Cytochrome p-450 polymorphisms and response to clopidogrel. New England Journal of Medicine. 2009;360(4):354-62.

10. Wang Y, Zhao X, Lin J, Li H, Johnston SC, Lin Y, et al. Association between CYP2C19 loss-of-function allele status and efficacy of clopidogrel for risk reduction among patients with minor stroke or transient ischemic attack. Jama. 2016;316(1):70-8.

11. Shi X, Ding M, Dong Z, Chen F, Ye J, Wang S, et al. Antioxidant properties of aspirin: characterization of the ability of aspirin to inhibit silica-induced lipid peroxidation, DNA damage, NF-κB activation, and TNF-α production. Molecular and cellular biochemistry. 1999;199(1-2):93-102.

12. Association AD. Cardiovascular disease and risk management: standards of medical care in diabetes—2018. Diabetes care. 2018;41(Supplement 1):S86-S104.

---

## [Decision Letter · Decision Letter 1]

15 Aug 2019

Aspirin better than clopidogrel on major adverse cardiovascular events reduction after ischemic stroke: a retrospective nationwide cohort study

PONE-D-19-17386R1

Dear Dr. Hu,

We are pleased to inform you that your manuscript has been judged scientifically suitable for publication and will be formally accepted for publication once it complies with all outstanding technical requirements.

With kind regards,

Aristeidis H. Katsanos, MD, PhD

Academic Editor

PLOS ONE

Additional Editor Comments (optional):

Reviewers' comments:

Reviewer's Responses to Questions

**Comments to the Author**

1. If the authors have adequately addressed your comments raised in a previous round of review and you feel that this manuscript is now acceptable for publication, you may indicate that here to bypass the “Comments to the Author” section, enter your conflict of interest statement in the “Confidential to Editor” section, and submit your "Accept" recommendation.

Reviewer #1: All comments have been addressed

2. Is the manuscript technically sound, and do the data support the conclusions?

Reviewer #1: Yes

3. Has the statistical analysis been performed appropriately and rigorously? 

Reviewer #1: Yes

4. Have the authors made all data underlying the findings in their manuscript fully available?

Reviewer #1: Yes

5. Is the manuscript presented in an intelligible fashion and written in standard English?

Reviewer #1: Yes

6. Review Comments to the Author

Reviewer #1: (No Response)

7. PLOS authors have the option to publish the peer review history of their article (what does this mean?). If published, this will include your full peer review and any attached files.

Reviewer #1: No

---

## [Editor Report · Acceptance letter]

20 Aug 2019

PONE-D-19-17386R1 

Aspirin better than clopidogrel on major adverse cardiovascular events reduction after ischemic stroke: a retrospective nationwide cohort study 

Dear Dr. Hu:

I am pleased to inform you that your manuscript has been deemed suitable for publication in PLOS ONE. Congratulations! Your manuscript is now with our production department. 

With kind regards,

on behalf of

Dr. Aristeidis H. Katsanos 

Academic Editor

PLOS ONE